# Beyond the Return: Off-policy Function Estimation under User-specified Error-measuring Distributions

**Audrey Huang**
Computer Science
University of Illinois at Urbana-Champaign
`audreyh5@illinois.edu`

**Nan Jiang**
Computer Science
University of Illinois at Urbana-Champaign
`nanjiang@illinois.edu`

## Abstract

Off-policy evaluation often refers to two related tasks: estimating the expected return of a policy and estimating its value function (or other functions of interest, such as density ratios). While recent works on marginalized importance sampling (MIS) show that the former can enjoy provable guarantees under realizable function approximation, the latter is only known to be feasible under much stronger assumptions such as prohibitively expressive discriminators. In this work, we provide guarantees for off-policy function estimation under only realizability, by imposing proper regularization on the MIS objectives. Compared to commonly used regularization in MIS, our regularizer is much more flexible and can account for an arbitrary user-specified distribution, under which the learned function will be close to the groundtruth. We provide *exact* characterization of the optimal dual solution that needs to be realized by the discriminator class, which determines the data-coverage assumption in the case of value-function learning. As another surprising observation, the regularizer can be altered to relax the data-coverage requirement, and completely eliminate it in the ideal case with strong side information.

## 1 Introduction

Off-policy evaluation (OPE) often refers to two related tasks in reinforcement learning (RL): estimating the expected return of a target policy using a dataset collected from a different *behavior* policy, versus estimating the policy's value function (or other functions of interest, such as density ratios). The former is crucial to hyperparameter tuning and verifying the performance of a policy before real-world deployment in offline RL [VLJY19; Pai+20; ZJ21]. The latter, on the other hand, plays an important role in (both online and offline) training, often as the subroutine of actor-critic-style algorithms [LP03; LSAB19], but is also generally more difficult than the former: if an accurate value function is available, one could easily estimate the return by plugging in the initial distribution.

Between the two tasks, the theoretical nature of off-policy return estimation is relatively well understood, especially in terms of the function-approximation assumptions needed for sample-complexity guarantees. Among the available algorithms, importance sampling (IS) and its variants [PSS00; TTG15; JL16] do not require any function approximation, but incur *exponential-in-horizon* variance; Fitted-Q Evaluation [EGW05; LVY19] can enjoy polynomial sample complexity under appropriate coverage assumptions, but the guarantee relies on the strong Bellman-completeness assumption on the function class; marginalized importance sampling (MIS) methods, which have gained significant attention recently [LLTZ18; XMW19; UHJ20; NCDL19], use two function classes to simultaneously approximate the value and the density-ratio (or weight) function and optimize minimax objectives. Notably, it is the only family of methods known to produce accurate return estimates with a polynomial sample complexity, when the function classes only satisfy the relatively weak realizability assumptions (i.e., they contain the true value and weight functions).

In comparison, little is known about off-policy function estimation, and the guarantees are generally less desirable. Not only do the limitations of IS and FQE on return estimation carry over to this more challenging task, but MIS also loses its major advantage over FQE: despite the somewhat misleading impression left by many prior works, that MIS can handle function estimation the same way as return estimation,[1] MIS for function estimation often requires unrealistic assumptions such as prohibitively expressive discriminators. For concreteness, a typical guarantee for function estimation from MIS looks like the following (see e.g., Theorem 4 of [LLTZ18] and Lemmas 1 and 3 of [UHJ20]):

**Proposition 1** (Function estimation guarantee for MIS, informal). *Suppose the offline data distribution $d^D$ satisfies $d^D(s,a) > 0, \forall s, a$. Given value-function class $\mathcal{Q}$ with $q^\pi \in \mathcal{Q}$ and weight class $\mathcal{W} = \mathbb{R}^{\mathcal{S} \times \mathcal{A}}$, $q^\pi = \arg\min_{q \in \mathcal{Q}} \max_{w \in \mathcal{W}} L(w, q)$ for some appropriate population loss function $L$.[2]*

To enable the identification of the value function $q^\pi$, the result requires the discriminator class $\mathcal{W}$ to be the *space of all possible functions over the state-action space* ($\mathcal{W} = \mathbb{R}^{\mathcal{S} \times \mathcal{A}}$). In the finite-sample regime, using such a class incurs a sample complexity that depends on the size of the state-action space, which completely defeats the purpose of function approximation.

In addition, these results only hold asymptotically, where the function of interest can be exactly identified in a point-wise manner. Such an overly strong guarantee is unrealistic in the finite-sample regime, where one can only hope to approximate the function well in an average sense *under some distribution*, i.e., finite-sample performance guarantees should ideally bound $\|\widehat{q} - q^\pi\|_{2,\nu}$ for the learned $\widehat{q}$, where $\|\cdot\|_{2,\nu}$ is $\nu$-weighted 2-norm. Such fine-grained analyses are non-existent in MIS. Even in the broader literature, such results not only require Bellman-completeness-type assumptions [UIJKSX21], they also come with some fixed $\nu$ (which is not necessarily $d^D$; see Section 2) and the user has no freedom in choosing $\nu$. This creates a gap in the literature, as downstream learning algorithms that use off-policy function estimation as a subroutine often assume the estimation to be accurate under specific distributions [KL02; AYBBLSW19] (see details in Appendix A).

To summarize, below are two important open problems on off-policy function estimation:

1. *Is it possible to obtain polynomial[3] sample complexity for off-policy function estimation, using function classes that only satisfy realizability-type assumptions?*

2. *Can we specify a distribution $\nu$ to the estimation algorithm, such that the learned function will be close to the groundtruth under $\nu$?*

In this work, we answer both open questions in the positive. By imposing proper regularization on the MIS objectives, we provide off-policy function estimation guarantees under only realizability assumptions on the function classes. Compared to commonly used regularization in MIS [NCDL19; ND20; YNDLS20], our regularizer is much more flexible and can account for an arbitrary user-specified distribution $\nu$, under which the learned function will be close to the groundtruth. We provide *exact* characterization of the optimal dual solution that needs to be realized by the discriminator, which determines the data-coverage assumption in value-function learning. As another surprising observation, the regularizer can be altered to relax the data-coverage requirement, and in the ideal case *completely eliminate* it when strong side information is available. Proof-of-concept experiments are also conducted to validate our theoretical predictions.

## 2 Related Works

**Regularization in MIS** The use of regularization is very common in the MIS literature, especially in DICE algorithms [NCDL19; NDKCLS19; YNDLS20]. However, most prior works that consider regularization use tabular derivations and seldom provide finite-sample function-approximation guarantees on even return estimation, let alone function estimation. (An exception is the work of [UIJKSX21], who analyze related estimators under Bellman-completeness-type assumptions; see the next paragraph.) More importantly, prior works provide very limited understanding in how choice of regularization affects learning guarantees, and have considered only naïve forms of regularization (state-action-*independent* and typically under $d^D$), under which different forms of regularization are essentially treated equally under a coarse-grained theory [YNDLS20]. In contrast, we provide much

---

[1]For example, [LSAB19] assumed a weight estimation oracle and cited [LLTZ18] as a possible instance.

[2]Concretely, one can choose $L$ as Eq.(4) without the regularization term, which recovers MQL in [UHJ20].

[3]By "polynomial", we mean polynomial in the horizon, the statistical capacities and the boundedness of the function classes, and the parameter that measures the degree of data coverage.

more fine-grained characterization of the effects of regularization, which leads to novel insights about how to design better regularizers, and existing DICE estimators are subsumed as special cases of our method when we choose very simple regularizers (see Remark 3 in Section 5).

**Fitted-Q Evaluation (FQE)** Outside the MIS literature, one can obtain return *and* value-function estimation guarantees via FQE [DJW20; CJ19; LVY19; UIJKSX21]. However, it is well understood that FQE and related approaches require Bellman-completeness-type assumptions, such as the function class being *closed* under the Bellman operator. Even putting aside the difference between completeness vs. realizability, we allow for a user-specified error-measuring distribution, which is not available in FQE or any other existing method. The only distribution these methods are aware of is the data distribution $d^D$, and even so, FQE and variants rarely provide guarantees on $\|\widehat{q} - q^\pi\|_{2,d^D}$, but often on the Bellman error (e.g., $\|\widehat{q} - \mathcal{T}^\pi \widehat{q}\|_{2,d^D}$) instead [UIJKSX21], and obtaining guarantees on a distribution of interest often requires multiple indirect translations and loose relaxations.

**LSTDQ** Our analyses focus on general function approximation. When restricted to linear classes, function estimation guarantees for $q^\pi$ under $d^D$ can be obtained by LSTDQ methods [LP03; BY09; DNP14] when the function class only satisfies realizability of $q^\pi$ [PKBK22]. However, this requires an additional matrix invertibility condition (see Assumption 3 of [PKBK22]), and it is still unclear what this condition corresponds to in general function approximation.[4] Moreover, many general methods—including MIS [UHJ20] and other minimax methods [ASM08; XCJMA21]—coincide with LSTDQ in the linear case, so the aforementioned results can be viewed as a specialized analysis leveraging the properties of linear classes.

**PRO-RL [ZHHJL22]** Our key proof techniques are adapted from [ZHHJL22], whose goal is offline policy learning. They learn the importance weight function $w^\pi$ for a near-optimal $\pi$, and provide $\|\widehat{w} - w^\pi\|_{2,d^D}$ guarantees as an intermediate result. Despite using similar technical tools, our most interesting and surprising results are in the value-function estimation setting, which is not considered by [ZHHJL22]. Our novel algorithmic insights, such as incorporating error-measuring distributions and approximate models in the regularizers, are also potentially useful in [ZHHJL22]'s policy learning setting. Our analyses also reveal a number of important differences between OPE and offline policy learning, which will be discussed in Appendix A.

## 3 Preliminaries

We consider off-policy evaluation (OPE) in Markov Decision Processes (MDPs). An MDP is specified by its state space $\mathcal{S}$, action space $\mathcal{A}$, transition dynamics $P : \mathcal{S} \times \mathcal{A} \to \Delta(\mathcal{S})$ ($\Delta(\cdot)$ is the probability simplex), reward function $R : \mathcal{S} \times \mathcal{A} \to \Delta([0,1])$, discount factor $\gamma \in [0,1)$, and an initial state distribution $\mu_0 \in \Delta(\mathcal{S})$. We assume $\mathcal{S}$ and $\mathcal{A}$ are finite and discrete, but their cardinalities can be arbitrarily large. Given a target policy $\pi : \mathcal{S} \to \Delta(\mathcal{A})$, a random trajectory $s_0, a_0, r_0, s_1, a_1, r_1, \ldots$ can be generated as $s_0 \sim \mu_0, a_t \sim \pi(\cdot|s_t), r_t \sim R(\cdot|s_t, a_t), s_{t+1} \sim P(\cdot|s_t, a_t), \forall t \geq 0$; we use $\mathbb{E}_\pi$ and $\mathbb{P}_\pi$ to refer to expectation and probability under such a distribution. The expected discounted return (or simply return) of $\pi$ is $J(\pi) := \mathbb{E}_\pi[\sum_t \gamma^t r_t]$. The Q-value function of $\pi$ is the unique solution of the Bellman equations $q^\pi = \mathcal{T}^\pi q^\pi$, with the Bellman operator $\mathcal{T}^\pi : \mathbb{R}^{\mathcal{S} \times \mathcal{A}} \to \mathbb{R}^{\mathcal{S} \times \mathcal{A}}$ defined as $\forall q \in \mathbb{R}^{\mathcal{S} \times \mathcal{A}}, (\mathcal{T}^\pi q)(s,a) := \mathbb{E}_{r \sim R(\cdot|s,a)}[r] + \gamma(P^\pi q)(s,a)$. Here $P^\pi \in \mathbb{R}^{|\mathcal{S} \times \mathcal{A}| \times |\mathcal{S} \times \mathcal{A}|}$ is the state-action transition operator of $\pi$, defined as $(P^\pi q)(s,a) := \mathbb{E}_{s' \sim P(\cdot|s,a), a' \sim \pi(\cdot|s')}[q(s',a')]$. Functions over $\mathcal{S} \times \mathcal{A}$ (such as $q$) are also treated as $|\mathcal{S} \times \mathcal{A}|$-dimensional vectors interchangeably.

In OPE, we want to estimate $q^\pi$ and other functions of interest based on a historical dataset collected by a possibly different policy. As a standard simplification, we assume that the offline dataset consisting of $n$ i.i.d. tuples $\{(s_i, a_i, r_i, s_i')\}_{i=1}^n$ sampled as $(s_i, a_i) \sim d^D, r \sim R(\cdot|s_i, a_i)$, and $s_i' \sim P(\cdot|s_i, a_i)$. We call $d^D$ the (offline) data distribution. As another function of interest, the (marginalized importance) weight function $w^\pi$ is defined as $w^\pi(s,a) := d^\pi(s,a)/d^D(s,a)$, where $d^\pi(s,a) = (1-\gamma)\sum_{t=0}^\infty \gamma^t \mathbb{P}_\pi[s_t = s, a_t = a]$ is the discounted state-action occupancy of $\pi$. For technical convenience we assume $d^D(s,a) > 0 \ \forall s, a$, so that quantities like $w^\pi$ are always well defined and finite.[5] Similarly to $q^\pi$, $w^\pi$ also satisfies a recursive equation, inherited from the Bellman

---

[4]It is hinted by [UHJ20] that the invertibility is related to a loss minimization condition in MIS, but the connection only holds for return estimation.

[5]It will be trivial to remove this assumption at the cost of cumbersome derivations. Also, these density ratios can still take prohibitively large values even if they are finite, and we will need to make additional boundedness assumptions to enable finite-sample guarantees anyway, so their finiteness does not trivialize the analyses.

flow equation for $d^\pi$: $d^\pi = (1-\gamma)\mu_0^\pi + \gamma \widetilde{P}^\pi d^\pi$, where $(s,a) \sim \mu_0^\pi \Leftrightarrow s \sim \mu_0, a \sim \pi(\cdot|s)$ is the initial state-action distribution, and $\widetilde{P}^\pi := (P^\pi)^\top$ is the transpose of the transition matrix.

**Function Approximation** We will use function classes $\mathcal{Q}$ and $\mathcal{W}$ to approximate $q^\pi$ and $w^\pi$, respectively. We assume finite $\mathcal{Q}$ and $\mathcal{W}$, and extension to infinite classes under appropriate complexity measures (e.g., covering number) is provided in Appendix G.

**Additional Notation** $\|\cdot\|_{2,\nu} := \sqrt{\mathbb{E}_\nu[(\cdot)^2]}$ is the weighted 2-norm of a function under distribution $\nu$. We also use a standard shorthand $f(s,\pi) := \mathbb{E}_{a \sim \pi(\cdot|s)}[q(s,a)]$. Elementwise multiplication between two vectors $u$ and $v$ of the same dimension is $u \circ v$, and elementwise division is $u/v$.

# 4 Value-function Estimation

In this section we show how to estimate $\widehat{q} \approx q^\pi$ with guarantees on $\|\widehat{q} - q^\pi\|_{2,\nu}$ for a user-specified $\nu$, and identify the assumptions under which provable sample-complexity guarantees can be obtained. We begin with the familiar Bellman equations, that $q^\pi$ is the unique solution to:

$$\mathbb{E}_{r \sim R(\cdot|s,a)}[r] + \gamma \mathbb{E}_{s' \sim P(\cdot|s,a)}[q(s',\pi)] - q(s,a) = 0, \ \forall s,a \in \mathcal{S} \times \mathcal{A}. \tag{1}$$

While the above set of equations uniquely determines $q = q^\pi$, this is only true if we can enforce *all* the $|\mathcal{S} \times \mathcal{A}|$ constraints, which is intractable in large state-space problems. In fact, even estimating (a candidate $q$'s violation of) a single constraint is infeasible as that requires sampling from the same state multiple times, which is related to the infamous double-sampling problem [Bai95].

To overcome this challenge, prior MIS works often relax Eq.(1) by taking a *weighted combination* of these equations, e.g.,

$$\mathbb{E}_{d^D}[w(s,a)(r(s,a) + \gamma q(s',\pi) - q(s,a))] = 0, \ \forall w \in \mathcal{W}. \tag{2}$$

Instead of enforcing $|\mathcal{S} \times \mathcal{A}|$ equations, we only enforce their linear combinations; the linear coefficients are $d^D(s,a) \cdot w(s,a)$, and $w$ belongs to a class $\mathcal{W}$ with limited statistical capacity to enable sample-efficient estimation. While each constraint in Eq.(2) can now be efficiently checked on data, this comes with a big cost that a solution to Eq.(2) is not necessarily $q^\pi$. Prior works handle this dilemma by aiming lower: instead of learning $\widehat{q} \approx q^\pi$, they only learn $\widehat{q}$ that can approximate the policy's return, i.e. $\mathbb{E}_{s \sim \mu_0}[\widehat{q}(s,\pi)] \approx J(\pi) = \mathbb{E}_{s \sim \mu_0}[q^\pi(s,\pi)]$. While [UHJ20] show that the latter is possible when $w^\pi \in \mathcal{W}$, they also show explicit counterexamples where $\widehat{q} \neq q^\pi$ even with infinite data. As a result, how to estimate $\widehat{q} \approx q^\pi$ under comparable realizability assumptions (instead of the prohibitive $\mathcal{W} = \mathbb{R}^{\mathcal{S} \times \mathcal{A}}$ as in Proposition 1) is still an open problem.

## 4.1 Estimator

We now describe our approach to solving this problem. Recall that the goal is to obtain error bounds for $\|\widehat{q} - q^\pi\|_{2,\nu}$ for some distribution $\nu \in \Delta(\mathcal{S} \times \mathcal{A})$ specified by the user. Note that we do *not* require information about $r$ and $s'$ that are generated after $(s,a) \sim \nu$ and only care about the $(s,a)$ marginal itself, so the user can pick $\nu$ in an arbitrary manner without knowing the transition and the reward functions of the MDP. We assume that $\nu$ is given in a way that we can take its expectation $\mathbb{E}_{(s,a) \sim \nu}[(\cdot)]$, and extension to the case where $\nu$ is given via samples is straightforward.

To achieve this goal, we first turn Eq.(1) into an equivalent *constrained convex program*: given a collection of *strongly convex* and differentiable functions $\{f_{s,a} : \mathbb{R} \to \mathbb{R}\}_{s,a}$—we will refer to the collection as $f$ and discuss its choice later—consider

$$\min_q \mathbb{E}_{(s,a) \sim \nu}[f_{s,a}(q(s,a))] \tag{3}$$
$$\text{s.t. } \mathbb{E}_{r \sim R(\cdot|s,a)}[r] + \gamma \mathbb{E}_{s' \sim P(\cdot|s,a)}[q(s',\pi)] - q(s,a) = 0, \ \forall s,a \in \mathcal{S} \times \mathcal{A}.$$

The constraints here are the same as Eq.(1). Since Eq.(1) uniquely determines $q = q^\pi$, the feasible space of Eq.(3) is a *singleton*, so we can impose any objective function on top of these constraints (here we use $\mathbb{E}_{(s,a) \sim \nu}[f_{s,a}(q(s,a))]$) and it will not change the optimal solution (which is always $q^\pi$, the only feasible point). As we will see, however, $f = \{f_{s,a} : \mathbb{R} \to \mathbb{R}\}_{s,a}$ will serve as an important regularizer in the function-approximation setting and is crucial to our estimation guarantees.

**Remark 1** ($(s,a)$-dependence of $f$)**.** Regularizers in prior works are $(s,a)$-*independent* [NCDL19; YNDLS20; ZHHJL22]. As we will see in Section 4.3, allowing for $(s,a)$-dependence is very important for designing regularizers with improved guarantees and performances.

We now rewrite (3) in its Lagrangian form, with $d^D \circ w$ serving the role of dual variables:

$$\min_q \max_w L_f^q(q, w) := \mathbb{E}_\nu[f_{s,a}(q(s,a))] + \mathbb{E}_{d^D}\left[w(s,a)\left(r(s,a) + \gamma q(s', \pi) - q(s,a)\right)\right]. \quad (4)$$

Finally, our actual estimator approximates Eq.(4) via finite-sample approximation of the population loss $L_f^q$, and searches over restricted function classes $\mathcal{Q}$ and $\mathcal{W}$ for $q$ and $w$, respectively:

$$\widehat{q} = \arg\min_{q \in \mathcal{Q}} \max_{w \in \mathcal{W}} \widehat{L}_f^q(q, w), \quad (5)$$

where $\widehat{L}_f^q(q,w) = \mathbb{E}_\nu[f_{s,a}(q(s,a))] + \frac{1}{n}\sum_{i=1}^n w(s_i, a_i)\left(r_i + \gamma q(s_i', \pi) - q(s_i, a_i)\right)$.

**Intuition for identification**   Before giving the detailed finite-sample analysis, we provide some high-level intuitions for why we can obtain the desired guarantee on $\|\widehat{q} - q^\pi\|_{2,\nu}$. Note that Eq.(4) is structurally similar to Eq.(2), and we still cannot verify the Bellman equation for $q^\pi$ in a per-state-action manner, so the caveat of Eq.(2) seems to remain; why can we identify $q^\pi$ under $\nu$?

The key here is to show that it suffices to check the loss function $L_f^q$ only under a special choice of $w$ (as opposed to all of $\mathbb{R}^{\mathcal{S} \times \mathcal{A}}$). Importantly, this special $w$ is *not* $w = w^\pi$;[6] rather, it is the saddle point of our regularized objective $L_f^q$: let $(q^\pi, w_f^*)$ be a saddle point of $L_f^q$ (we will give the closed form of $w_f^*$ later). As long as $w_f^* \in \mathcal{W}$—even if $\mathcal{W}$ is extremely "simple" and contains nothing but $w_f^*$—we can identify $q^\pi$.

To see that, it is instructive to consider the special case of $\mathcal{W} = \{w_f^*\}$ and the limit of infinite data. In this case, our estimator becomes $\arg\min_{q \in \mathcal{Q}} L_f^q(q, w_f^*)$. By the definition of saddle point:

$$L_f^q(q^\pi, w_f^*) \leq L_f^q(q, w_f^*), \ \forall q.$$

While this shows that $q^\pi$ is a minimizer of the loss, it does not imply that it is a unique minimizer. However, identification immediately follows from the convexity brought by regularization: since $f : \mathbb{R} \to \mathbb{R}$ is strongly convex, $q \to \mathbb{E}_\nu[f_{s,a}(q(s,a))]$ as a mapping from $\mathbb{R}^{\mathcal{S} \times \mathcal{A}}$ to $\mathbb{R}$ is strongly convex under $\|\cdot\|_{2,\nu}$ (see Lemma 7 in Appendix B for a formal statement and proof), and $L_f^q(q, w_f^*)$ inherits such convexity since the other terms are affine in $q$. It is then obvious that $q^\pi$ is the unique minimizer of $L_f^q(q^\pi, w_f^*)$ up to $\|\cdot\|_{2,\nu}$, that is, any minimizer of $L_f^q$ must agree with $q^\pi$ on $(s,a)$ pairs supported on $\nu$. Our finite-sample analysis below shows that the above reasoning is robust to finite-sample errors and the inclusion of functions other than $w_f^*$ in $\mathcal{W}$.

## 4.2   Finite-sample Guarantees

In this subsection we state the formal guarantee of our estimator for $q^\pi$ and the assumptions under which the guarantee holds. We start with the condition on the regularization function $f$:

**Assumption 1** (Strong convexity of $f$). Assume $f_{s,a} : \mathbb{R} \to \mathbb{R}$ is nonnegative, differentiable, and $M^q$-strongly convex for each $s \in \mathcal{S}, a \in \mathcal{A}$. In addition, assume both $f_{s,a}$ and its derivative $f_{s,a}'$ take finite values for any finite input.

This assumption can be concretely satisfied by a simple choice of $f_{s,a}(x) = \frac{1}{2}x^2$, which is independent of $(s,a)$ and yields $M^q = 1$. Alternative choices of $f$ will be discussed in Section 4.3. Next are the realizability and boundedness of $\mathcal{W}$ and $\mathcal{Q}$:

**Assumption 2** (Realizability). Suppose $w_f^* \in \mathcal{W}$, $q^\pi \in \mathcal{Q}$.

**Assumption 3** (Boundedness of $\mathcal{W}$ and $\mathcal{Q}$). Suppose $\mathcal{W}$ and $\mathcal{Q}$ are bounded, that is, $C_{\mathcal{Q}}^q := \max_{q \in \mathcal{Q}} \|q\|_\infty < \infty$, $\quad C_{\mathcal{W}}^q := \max_{w \in \mathcal{W}} \|w\|_\infty < \infty$.

As a remark, Assumption 2 implicitly assumes the existence of $w_f^*$. As we will see in Section 4.3, the existence and finiteness of $w_f^*$ is automatically guaranteed given the finiteness of $f_{s,a}'$ (Assumption 1) and $d^D(s,a) > 0 \ \forall s, a$. More importantly, Assumptions 2 and 3 together imply that $\|q^\pi\|_\infty \leq C_{\mathcal{Q}}^q$ and $\|w_f^*\|_\infty \leq C_{\mathcal{W}}^q$, which puts constraints on how small $C_{\mathcal{Q}}^q$ and $C_{\mathcal{W}}^q$ can be. For example, it is common to assume that $C_{\mathcal{Q}}^q = \frac{1}{1-\gamma}$, i.e., the maximum possible return when rewards are bounded

---

[6]In fact, $w^\pi$ should not appear in our analysis at all: $w^\pi$ is defined w.r.t. the initial distribution of the MDP, $\mu_0$, which has nothing to do with our goal of bounding $\|\widehat{q} - q^\pi\|_{2,\nu}$.

in $[0, 1]$, and this way $\|q^\pi\|_\infty \leq C_Q^q$ will hold automatically. The magnitude of $\|w_f^*\|_\infty$ and $C_W^q$, however, is more nuanced and interesting, and we defer the discussion to Section 4.3.

Now we are ready to state the main guarantee for identifying $q^\pi$. All proofs of this section can be found in Appendix B.

**Theorem 2.** *Suppose Assumptions 1, 2, 3 hold. Then, with probability at least $1 - \delta$,*

$$\|\widehat{q} - q^\pi\|_{2,\nu} \leq 2\sqrt{\frac{\epsilon_{stat}^q}{M^q}},$$

*where $\epsilon_{stat}^q = \left(C_W^q + (1+\gamma)C_W^q C_Q^q\right)\sqrt{\frac{2\log\frac{2|\mathcal{W}||\mathcal{Q}|}{\delta}}{n}}$.*

Theorem 2 shows the desired bound on $\|\widehat{q} - q^\pi\|_{2,\nu}$, which depends on the magnitude of functions in $\mathcal{W}$ and $\mathcal{Q}$ as well as their logarithmic cardinalities, which are standard measures of statistical complexity for finite classes. One notable weakness is the $O(n^{-1/4})$ slow rate; this is due to translating the $\epsilon_{stat}^q = O(n^{-1/2})$ deviation between $L$ and $\widehat{L}$ into $\|\widehat{q} - q^\pi\|_{2,\nu}$ via a convexity argument, which takes square root of the error. The possibility of and obstacles to obtaining an $O(n^{-1/2})$ rate will be discussed in Section 7.

### 4.3 On the Closed Form of $w_f^*$ and the Data Coverage Assumptions

One unusual aspect of our guarantees in Section 4.2 is that we do not make any explicit data coverage assumptions, yet such assumptions are known to be necessary even for return estimation (typically the boundedness of $w^\pi = d^\pi/d^D$). Indeed, our data-coverage assumption is implicit in Assumptions 2 and 3, which require $\|w_f^*\|_\infty \leq C_W^q < \infty$. If data fails to provide sufficient coverage, $\|w_f^*\|_\infty$ will be large and our bound in Theorem 2 will suffer due to a large value of $C_W^q$.

To make the data coverage assumption explicit, we provide the closed-form expression of $w_f^*$:

**Lemma 3.** *The saddle point of (4) is $(q^\pi, w_f^*) = \arg\min_q \arg\max_w L_f^q(q, w)$, where*

$$w_f^* = (I - \gamma\widetilde{P}^\pi)^{-1}\left(\nu \circ f'(q^\pi)\right)/d^D. \tag{6}$$

*Here $f'(q^\pi)$ is the shorthand for $[f'_{s,a}(q^\pi(s,a))]_{s,a} \in \mathbb{R}^{\mathcal{S}\times\mathcal{A}}$.*

The closed-form expression in Eq.(6) looks very much like a density ratio: if we replace $\nu \circ f'(q^\pi)$ with $\mu_0^\pi$, we have $(I - \gamma\widetilde{P}^\pi)^{-1}\mu_0^\pi = d^\pi/(1-\gamma)$, and the expression would be the ratio between $d^\pi$ and $d^D$ (up to a horizon factor). Therefore, $w_f^*$ can be viewed as the density ratio of $\pi$ against $d^D$ when $\pi$ starts from the "fake" initial distribution $\nu \circ f'(q^\pi)$. However, $\nu \circ f'(q^\pi)$ is in general not a valid distribution, as it is not necessarily normalized or even non-negative, making $\|w_f^*\|_\infty$ difficult to intuit. Below we give relaxations of $\|w_f^*\|_\infty$, which are more interpretable and give novel insights into how to relax the data-coverage assumption via tweaking $f$.

**Proposition 4.** $\|w_f^*\|_\infty \leq \frac{1}{1-\gamma} \cdot \|d_\nu^\pi/d^D\|_\infty \cdot \|f'(q^\pi)\|_\infty$, *where $d_\nu^\pi$ is the discounted state-action occupancy of $\pi$ under $\nu$ as the initial state-action distribution.*

The proposition states that $\|w_f^*\|_\infty$ can be bounded if data provides sufficient coverage over $d_\nu^\pi$, and if $f'(q^\pi)$ is bounded. The former shows that $d^D$ needs to cover not only $\nu$, but also state-action pairs reachable by $\pi$ starting from $\nu$. The latter is easily satisfied, and can be bounded again for concrete choices of $f$, e.g. $\|f'(q^\pi)\|_\infty \leq \|q^\pi\|_\infty \leq \frac{1}{1-\gamma}$ for $f_{s,a}(x) = \frac{1}{2}x^2$.

**Designing $f$ to relax the coverage assumption** Lemma 3 shows that the coverage assumption (bounded $\|w_f^*\|_\infty$) depends on $f$ (or rather its derivative $f'$), which opens up the possibility of properly designing $f$ to relax it. In fact, we could completely eliminate the coverage assumption if we could set $f'(q^\pi) = \mathbf{0}$, but that would require unrealistically strong side information.

As a concrete example, consider $f_{s,a}(x) = \frac{1}{2}(x - q^\pi(s,a))^2$, and it is easy to verify that $f'_{s,a}(q^\pi(s,a)) = x - q^\pi(s,a)|_{x=q^\pi(s,a)} = 0$. Compared to $f_{s,a}(x) = \frac{1}{2}x^2$, the new $f$ essentially adds a 1st-order term $q^\pi(s,a) \cdot x$ to change $w_f^*$, while leaving the convexity required by Assumption 1 intact, which only depends on the 2nd-order term $\frac{1}{2}x^2$. Of course, this is not a viable choice of $f$ in practice as it requires knowledge of $q^\pi$, which is precisely our learning target.

While the reason $f_{s,a}(x) = \frac{1}{2}(x - q^\pi(s,a))^2$ can eliminate the coverage requirement is obvious retrospectively ($q^\pi$ already minimizes $\mathbb{E}_\nu[f(q)]$ even without any data), our analyses apply much more generally and characterize the effects of arbitrary $f$ on the coverage assumption. Inspired by this example, we can consider practically feasible choices such as $f_{s,a}(x) = \frac{1}{2}(x - \widetilde{q}(s,a))^2$, where $\widetilde{q}$ is an approximation of $q^\pi$ obtained by other means, e.g. a guess based on domain knowledge. If $\widetilde{q} \approx q^\pi$, our estimator enjoys significantly relaxed coverage requirements. But even if $\widetilde{q}$ is a poor approximation of $q^\pi$, it does not affect our estimation guarantees as long as the condition implied by Proposition 4 is satisfied. (In fact, $f_{s,a}(x) = \frac{1}{2}x^2$ is a special case of $\widetilde{q} \equiv 0$.) Such a use of approximate models is similar to how doubly robust estimators [DLL11; JL16; TB16] enjoy reduced variance given an accurate model, and remain unbiased even if the approximate model is arbitrarily poor. We will show in Section 6 that this idea is empirically effective.

## 5  Weight-function Estimation

Similar to value-function estimation, our methodology can also be applied to estimate the weight function $w^\pi$. Due to the similarity with Section 4 in the high-level spirit, we will be concise in this section and only explain in detail when there is a conceptual difference from Section 4. Some notations (such as the function classes $\mathcal{W}$ and $\mathcal{Q}$) will be abused, but we emphasize that this section considers a different learning task than Section 4, so they should be viewed as different objects (e.g., the realizability assumptions for $\mathcal{W}$ and $\mathcal{Q}$ below will be different from those in Section 4).

As before, we assume that the user provides a distribution[7] $\eta \in \Delta(\mathcal{S} \times \mathcal{A})$ and our goal is to develop an estimator with guarantees on $\|\widehat{w} - w^\pi\|_{2,\eta}$. Analogous to Section 4, consider

$$\min_w \mathbb{E}_{(s,a) \sim \eta}[f_{s,a}(w(s,a))] \tag{7}$$
$$\text{s.t. } d^D(s,a)w(s,a) = (1-\gamma)\mu_0^\pi(s,a) + \gamma \sum_{s',a'} P^\pi(s,a|s',a')d^D(s',a')w(s,a), \ \forall s,a.$$

Here $f = \{f_{s,a}\}_{s,a}$ will need to satisfy similar assumptions as in Section 4. The constraints are the Bellman flow equations with a change of variable $d(s,a) = d^D(s,a) \cdot w(s,a)$. Their unique solution is $d(s,a) = d^\pi(s,a)$ (and hence $w(s,a) = d^\pi(s,a)/d^D(s,a)$), thus the feasible space is again a singleton, and the objective does not alter the optimal solution. We then use dual variables $q$ to rewrite (7) in its Lagrangian form: $\min_w \max_q L_f^w(q,w) :=$

$$\mathbb{E}_\eta[f_{s,a}(w(s,a))] + (1-\gamma)\mathbb{E}_{\mu_0}[q(s,\pi)] + \mathbb{E}_{d^D}[w(s,a)(\gamma q(s',\pi) - q(s,a))] \tag{8}$$

We approximate the saddle-point solutions by optimizing the empirical loss $\widehat{L}_f^w$ over restricted function classes $\mathcal{W}, \mathcal{Q}$: $\widehat{w} = \arg\min_{w \in \mathcal{W}} \max_{q \in \mathcal{Q}} \widehat{L}_f^w(q,w)$, where $\widehat{L}_f^w(q,w) := \mathbb{E}_\eta[f_{s,a}(w(s,a))] + (1-\gamma)\frac{1-\gamma}{n_0}\sum_{j=1}^{n_0} q(s_j,\pi) + \frac{1}{n}\sum_{i=1}^n w(s_i,a_i)(\gamma q(s_i',\pi) - q(s_i,a_i))$, and $\{s_j\}_{j=1}^{n_0}$ is a separate dataset sampled i.i.d. from $\mu_0$ to provide information about the initial distribution.

We provide the closed-form expression for the saddle point of $L_f^w$ below, which resembles the Q-function for a proxy reward function $f'(w^\pi) \circ \eta/d^D$.

**Lemma 5.** *The closed form solutions of* (8) *are* $(w^\pi, q_f^*) = \arg\min_w \arg\max_q L_f^w(q,w)$, *where*

$$q_f^* = (I - \gamma P^\pi)^{-1}(f'(w^\pi) \circ \eta/d^D). \tag{9}$$

**Remark 2** (Data Coverage Assumption). As we will see, the only data coverage assumption we need is the boundedness of $w^\pi = d^\pi/d^D$. Since $w^\pi$ is the function of interest and practical algorithms can only output functions of well-bounded ranges, such an assumption is an essential part of the learning task itself and hardly an additional requirement. Moreover, unlike Section 4, changing $f$ here will not affect the data-coverage assumption, though it still alters $q_f^*$, and a properly chosen $f$ (e.g., with $f'(w^\pi) \approx 0$) can still result in a $q_f^*$ with small magnitude and thus make learning easier.

**Remark 3** (Connection to DualDICE). We can recover DualDICE [NCDL19] by choosing $f_{s,a}(x) = \frac{1}{2}x^2$ and $\nu = d^D$. Despite producing the same estimator, the derivations and assumptions under which the two works analyze the estimator are different. Their Theorem 2 only provides return

---

[7]Recall we assume $d^D(s,a) > 0 \ \forall s,a$ for technical convenience. When this is not the case, $\eta$ should be supported on $d^D$, as the target function $w^\pi$ is only defined on the support of $d^D$.

estimation guarantees, and depends on an implicit assumption of highly expressive function classes[8] similar to Proposition 1. Moreover, they do not characterize how the choice of $f$ can affect the learning guarantees (their $f$ is $(s, a)$-independent). This is one of the main insights of our paper and leads to the discovery of more practical regularizers, e.g. $f_{s,a}(x) = \frac{1}{2}(x - \tilde{w}(s,a))^2$ with model $\tilde{w}$.

Below we present the assumptions, then learning guarantee for $\hat{w}$.

**Assumption 4** (Strongly Convex Objective). Suppose for all $s, a, f_{s,a}$ is differentiable, non-negative, and $M^w$-strongly convex. Further, suppose $f_{s,a}$ and its derivative take finite values on any finite inputs, and let $C_f^w := \max_{w \in \mathcal{W}} ||f(w)||_\infty$.

**Assumption 5** (Realizability). Suppose $w^\pi \in \mathcal{W}$, $q_f^* \in \mathcal{Q}$.

**Assumption 6** (Bounded $\mathcal{W}$ and $\mathcal{Q}$). Let $C_{\mathcal{W}}^w := \max_{w \in \mathcal{W}} ||w||_\infty$ and $C_{\mathcal{Q}}^w := \max_{q \in \mathcal{Q}} ||q||_\infty$. Suppose $\mathcal{W}$ and $\mathcal{Q}$ are bounded function classes, that is, $C_{\mathcal{W}}^w < \infty$ and $C_{\mathcal{Q}}^w < \infty$.

**Theorem 6.** *Suppose Assumptions 4, 5, 6 hold. Then w.p. $\geq 1 - \delta$, $||\hat{w} - w^\pi||_{2,\eta} \leq 2\sqrt{\frac{\epsilon_{stat}^w}{M^w}}$, where*

$$\epsilon_{stat}^w = \left(C_f^w + (1+\gamma)C_{\mathcal{W}}^w C_{\mathcal{Q}}^w\right) \sqrt{\frac{2\log\frac{4|\mathcal{Q}||\mathcal{W}|}{\delta}}{n}} + (1-\gamma)C_{\mathcal{Q}}^w \sqrt{\frac{2\log\frac{4|\mathcal{Q}|}{\delta}}{n_0}}.$$

## 6 Experiments

We now provide experimental results to verify our theoretical predictions and insights. As [YNDLS20] have performed extensive experiments on return estimation with simple regularization ($f_{s,a}(x) = \frac{1}{2}x^2$), we focus on the task of $q^\pi$ estimation, and the following two questions unique to our work:

**Q1.** When the goal is to minimize $||\hat{q} - q^\pi||_{2,\nu}$, how much benefit does regularizing with $\nu$ bring in practice, compared to regularizing with other distributions (or no regularization at all)?

**Q2.** Can incorporating (even relatively poor) models in regularization (e.g., $f_{s,a}(x) = \frac{1}{2}(x - \tilde{q}(s,a))^2$ from Section 4.3) improve estimation?

**Setup** We study these questions in a large tabular Gridwalk environment [NCDL19; YNDLS20], with a deterministic target policy $\pi$ that is optimal, and a behavior policy that provides limited coverage over the target policy; see Appendix D for further details. To mimic the identification challenges associated with restricted function classes, we use a linear function class $\mathcal{Q} = \{\Phi^\top \alpha : \alpha \in \mathbb{R}^d\}$ and discriminator class $\mathcal{W} = \{\tilde{\Phi}^\top \beta : \beta \in \mathbb{R}^k\}$, where $k < d \ll |\mathcal{S} \times \mathcal{A}|$. The features $\Phi \in \mathbb{R}^{|\mathcal{S} \times \mathcal{A}| \times d}$, $\tilde{\Phi} \in \mathbb{R}^{|\mathcal{S} \times \mathcal{A}| \times k}$ are chosen to satisfy the realizability assumptions of all estimators. Under linear classes, our estimator (Eq.(5)) becomes a convex optimization problem with $d$ variables and $k$ linear constraints, and can be solved by standard packages. This allows us to avoid difficult minimax optimization—which is still an open problem in the MIS literature—and focus on the statistical behaviors of our estimators, which is what our theoretical predictions are about.

**Remark 4.** When no regularization is used, our linear estimator coincides with MQL [UHJ20]. If we further had $\tilde{\Phi} = \Phi$, the estimator would coincide with LSTDQ. While Section 2 mentioned that LSTDQ enjoys function-estimation guarantees [PKBK22] (and folklore suggests they extend to $\tilde{\Phi} \neq \Phi$), the guarantee only holds in the regime of $k \geq d$, i.e., the $k$ linear constraints are *over-determined*. In our case, however, we have *under-determined* constraints ($k < d$), creating a more challenging learning task (which our theory can handle) where LSTDQ's guarantees do not apply.

**Choice of Distributions** We consider a set of diverse distributions $\mathcal{V} = \{d^D, \mu_0^\pi, d^\pi, U, p\}$, where $U$ is uniform over $\mathcal{S} \times \mathcal{A}$ and $p \propto (d^\pi \circ \mathbb{I}[w^\pi > 50])$. The distribution $p$ isolates the least-covered states reached by $\pi$, which makes learning an accurate Q-function on $\nu$ a harder task.

**Results for Q1** We use a default regularizer $f = \frac{1}{2}x^2$ with different regularizing distributions $\nu \in \mathcal{V}$, and measure $||\hat{q} - q^\pi||_{2,\nu'}$ for different $\nu'$. The results are shown in Figure 1, which exhibit the expected trend: for example, regularizing with $\nu = p$ performs poorly when the error is measured under $\nu' = d^D$ and $U$ due to the large mismatch between $\nu$ and $\nu'$. However, when $\nu' = p$ (rightmost panel), regularizing with $\nu = p$ significantly outperforms others. Similar behaviors can also be observed on $U$, though they are certainly not absolute (e.g., $\nu = d^D$ does not do very well on

---

[8]In our notation, they measure the approximation error of $\mathcal{W}$ as $\max_{w' \in \mathbb{R}^{\mathcal{S} \times \mathcal{A}}} \min_{w \in \mathcal{W}} ||w - w'||$, essentially requiring $\mathcal{W}$ (and similarly $\mathcal{Q}$) to closely approximate every function over $\mathcal{S} \times \mathcal{A}$. However, we suspect that they could have measured realizability errors instead without changing much of their proofs.

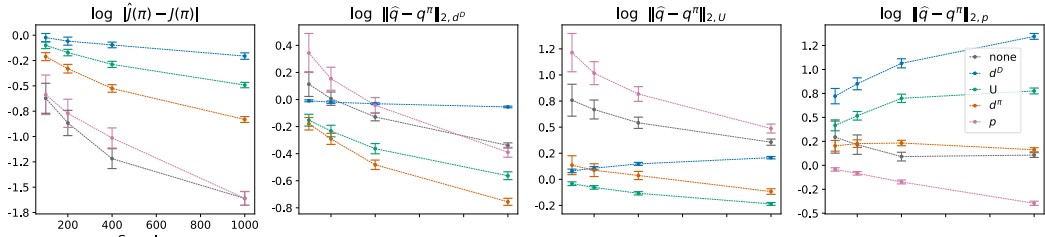

Figure 1: Error of off-policy return and function estimation as a function of sample size. Legend shows regularizing distribution $\nu$ and header shows error-measuring distribution $\nu'$ (see text). Error bars show 95% confidence intervals calculated from 1000 runs.

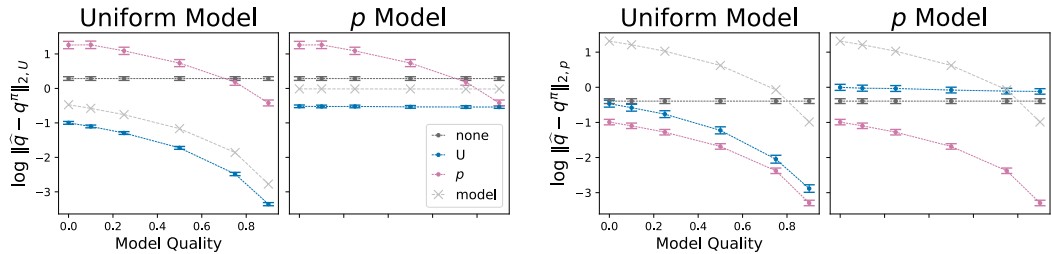

Figure 2: Estimation error when the regularizer incorporates a model $\widetilde{q}$, where x-axes represent the parameter $m$ that controls the quality of $\widetilde{q}$. The "model" line shows performance of $\widetilde{q}$. Sample size is 500 and the results are from 500 runs.

$\nu' = d^D$), which suggests potential directions for more refined and accurate theory. Moreover, using no regularization ("none") generally does not perform well for any $\nu'$, but still manages to achieve a high accuracy for return estimation $J(\pi)$, which is consistent with prior theory [e.g., UHJ20] that return estimation does not require regularization.

**Results for Q2** We now use $f_{s,a}(x) = \frac{1}{2}(x - \widetilde{q}(s,a))^2$ with different $\widetilde{q}$ to verify how the quality of $\widetilde{q}$ affect estimation accuracy. We first consider a "uniform model" $\widetilde{q} = mq^\pi + (1-m)\overline{q}$, where $\overline{q}$ is a constant and $m \in [0,1]$ controls the quality $\widetilde{q}$. As shown from Panels 1 & 3 in Figure 2, our estimator's accuracy generally improves with a better $\widetilde{q}$ (i.e., as $m$ increases). Moreover, equipping $\widetilde{q}$ with an appropriate regularizing distribution $\nu$ (e.g., $\nu = U$ for both panels) can significantly outperform no regularization, even with a very poor $\widetilde{q}$ (e.g., $m = 0.1$). It also outperforms the model prediction itself (i.e., $\widehat{q} = \widetilde{q}$), showing that the improvement is not due to our estimator simply taking predictions from $\widehat{q}$, but using the regularization to better assist the identification of $q^\pi$ from data.

The previous model's quality is uniform across $\mathcal{S} \times \mathcal{A}$. We then consider a scenario where $\widetilde{q}$ is zeroed out outside $p$'s support, making it only a good approximation of $q^\pi$ on $p$. In this case, we see that regularization cannot benefit much from the model when the error is measured on $\nu' = U$ (Panel 2), but when $\nu' = \nu = p$ (Panel 4), regularization can still bring benefits, as expected from our theory.

## 7 Discussion and Conclusion

In this paper we showed that proper regularization can yield function-estimation guarantees for MIS methods under only realizable function approximation. Compared to prior works, our regularizer is more flexible and can accommodate a user-specified error-measuring distribution. Further theoretical investigation provides fine-grained characterization of how the choice of regularization affects learning guarantees, which leads to the discovery of regularizers that incorporate approximate models (such as $\widetilde{q}$). While the superiority of such regularizers is perhaps obvious retrospectively, it is not allowed in the prior works' derivation that assumes $(s,a)$-independent regularization, and our theoretical results provide a deep understanding for even more general regularization schemes. In Appendix A, we provide further discussions on two topics: (1) the barriers to obtaining a faster $O(n^{-1/2})$ rate, and (2) comparison to [ZHHJL22] reveals interesting differences between off-policy function estimation and policy learning, and insights in this paper may also be useful for the policy learning task.

## Acknowledgments and Disclosure of Funding

The authors thank Jinglin Chen, Wenhao Zhan, and Jason Lee for valuable discussions during the early phase of the project, and Anonymous Reviewer e62P for insightful feedback that helped improve this paper during the review process. NJ acknowledges funding support from ARL Cooperative Agreement W911NF-17-2-0196, NSF IIS-2112471, NSF CAREER IIS-2141781, and Adobe Data Science Research Award.

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
