# OpenReview forum: "Beyond the Return: Off-policy Function Estimation under User-specified Error-measuring Distributions"
_NeurIPS.cc/2022/Conference — NeurIPS 2022 Accept_

### Official Review · Reviewer_LV55 · 2022-07-11

**Rating:** 6
**Confidence:** 3
**Soundness:** 3 good
**Presentation:** 3 good
**Contribution:** 3 good

**Summary:**

The paper studies estimating Q function in the offline setup. The state-action space is assumed to be discrete and the performance of the estimation is measured under a user-specified measure that can be different from the data distribution. The major caveat stems from the distribution shift and existing results requires completeness and realizability assumptions. This work proposes a Lagrangian method and provides statistical guarantees in the absent of completeness assumption. The analytical framework is further extended to weight function estimation.

**Questions:**

The Lagrangian form in Equation (4) is obtained from the convex optimization problem in (3). Do we gain any intuition from writing out this convex program? In fact, we can directly use $f_{s, a}$ as a regularizer and obtain Equation (4). On the other hand, there seems to be multiple ways to formulate a Lagrangian function, e.g., we can use a weighted quadratic penalty term $w(s, a) (r(s, a) +\gamma q(s', \pi) - q(s, a))^2$. Does the linear penalty in the Lagrangian form work better? Intuitively speaking, quadratic term is naturally related to penalized least square estimation.

How does Assumption 3 compare to concentrability conditions commonly used in RL literature, which assumes the ratio of visitation measure is bounded? In addition, [DJW20] uses a restricted chi-square divergence term to measure the distributional shift. They showed that the restricted chi-square divergence is tighter compared to density ratios. Can authors also comment on the connection between restricted chi-square divergence to $w_f^*$?

I am also curious about any connection to "Bellman-consistent Pessimism for Offline Reinforcement Learning", where an information theoretic confidence set of Q functions is constructed using offline data. The primary goal is different -- policy learning in the Bellman-consistent paper, however, they still obtain an estimation of Q function.


**Limitations:**

I don't see very obvious limitations in the paper, comparing to concurrent theoretical studies of RL. As mentioned, the assumption on weight function $w_f^*$ may be a bit strong.

The authors discussed the barriers of obtaining faster rate $n^{-1/2}$ in Appendix, which mainly owes to the regularity of the Lagrangian function. I am interested in whether the $n^{-1/2}$ rate is ever possible with novel techniques.

**Strengths And Weaknesses:**

Offline off-policy evaluation is a very important problem in RL and has many applications across various domains. The paper focuses on addressing the change of measure issue between evaluating and sampling distributions. Moreover, the paper only needs realizability assumption. Both of these advances are important in off-policy evaluation. There are many existing works studying off-policy evaluation and to my knowledge, the proposed method here is new.

The paper is relatively easy to follow, with some minor issues on notations. The result seems correct and sound, although I have not checked the detailed proofs.

I will discuss some weaknesses in the questions and limitations section.

---

> ### Author Response · Authors · 2022-07-28
> **Response**
>
> We thank the reviewer for their valuable comments, and respond to the questions below.
>
> ---
>
> - Intuition of Eq.(3) and (4):
>
> The key intuition in the convex program of (3) is that the feasible space is a singleton so any objective function wouldn’t change the solution (L173-174). One can certainly write down Eq.(4) directly without Eq.(3) but that is a bit heuristic.
>
> ---
>
> - Quadratic Penalty
>
> We believe the quadratic penalty doesn’t work, because $(r+\gamma q(s’, \pi) - q(s,a))^2$ runs into the classical double-sampling problem, that this loss includes an undesirable additional variance term due to the randomness of $s’|s,a$. In fact, MIS methods are, in some sense, designed to combat the double-sampling issue by taking *plain* average over the TD errors $r+\gamma q(s’, \pi) - q(s,a)$ (L151), and squaring the error beats that purpose.
>
> ---
>
> - Assumption 3 vs Concentrability
>
> There are a few different concentrability assumptions in offline RL: in policy learning setting, classical AVI/API analyses require “all-policy concentrability” $\max_\pi ||d^\pi/d^D||\_\infty$ bounded [CJ19]. For off-policy evaluation, this assumption can be weakened to “single-policy concentrability” on only the target policy $\pi$, i.e. $||d^\pi/d^D||\_\infty$ bounded. In comparison to the latter, our data assumption (Assumption 3, combined with the closed-form expression of $w_f^\star$; see Sec 4.3) is similar but can be more relaxed. For example, Prop 4 shows that $w_f^\star$ can be bounded if $d\_{\nu}^\pi/d^D$ is bounded, which matches “single-policy concentrability” when $\nu = \mu_0^\pi$. However, Prop 4 can be even weaker than this depending on the $\nu$, but more notably, when the regularizer $f$ is properly chosen (see extensive discussion in Sec 4.3).
>
> ---
>
> - Assumption 3  vs $\chi$-square divergence [DJW20]
>
> Great question! Indeed the $\chi$-square divergence is nice as it relaxes the concentrability coefficient by leveraging the structure of the function class. Unfortunately we can’t get this relaxation yet, and this is an open question that already existed in [ZHHJL22]. On the other hand, we also have a mechanism for relaxing our data assumption, namely via designing appropriate regularizers that yield small $f’(q^\pi)$ (Sec 4.3), which [DJW20] and related approaches do not enjoy.
>
> To see why it’s hard to get $\chi$-square divergence relaxation in our setting, one needs to look at a more “transparent” definition of [DJW20]’s divergence (*), e.g., Def. 1 from [XCJMA21] (“Bellman-consistent pessimism”). The definition measures how much the squared Bellman error $||f - \mathcal{T}^\pi f||\_{2, d^D} = \mathbb{E}\_{d^D}[(f-\mathcal{T}^\pi f)^2]$ will be amplified by distribution shift, thus is specific to approaches that estimate the squared Bellman errors (note the square inside expectation). Since squared Bellman error can only be estimated under Bellman-completeness-type assumptions (due to the double-sampling issue), such a divergence measure is incompatible with our approaches that only assume realizability.
>
> (*) To be clear, [DJW20]’s definition is technically correct, but they essentially started with the same definition as [XCJMA21] and relaxed & simplified it using other assumptions in their paper (Bellman completeness + linear function class), so the real structure of the divergence becomes difficult to recognize.
>
> ---
>
> - Connection to "Bellman-consistent Pessimism”
>
> You are right that [XCJMA21] also involves off-policy evaluation. First, they do *not* require function estimation, and return estimation is sufficient for them (which is what they guarantee). In fact, the term "Bellman-consistent Pessimism” exactly refers to the fact that they only need good uncertainty quantification for the return of each policy (which is done via maintaining a version space), which contrasts previous work based on bonus-based “pointwise pessimism” (e.g., Jin et al’21). Second, the way they form version space is similar to Antos et al. 08, which requires Bellman-completeness. Similar algorithms have also been analyzed in [CJ21,UIJKSX21], so the spirit of this work is covered by our discussion in L89.
>
> Antos et al. 2008. Learning near-optimal policies with Bellman-residual minimization based fitted policy iteration and a single sample path.

---

> > ### Comment · Reviewer_LV55 · 2022-08-09
> > **Thank you for your response**
> >
> > Thanks for the detailed response.
> >
> > I am satisfied with the response and would like to keep my score.

---

### Official Review · Reviewer_e62P · 2022-07-19

**Rating:** 6
**Confidence:** 4
**Soundness:** 4 excellent
**Presentation:** 3 good
**Contribution:** 2 fair

**Summary:**

The paper studies the problem of learning the Q-function and density ratio for a given policy in offline reinforcement learning settings, where we have IID state, action, reward, next state tuples collected from the stationary distribution of some behavioral policy. They argue that in past work the focus has been on guarantees on the accuracy of policy value evaluation using estimates of these functions, rather than guarantees on the accuracy of the estimates of these functions themselves. The authors propose a new more general framework for regularized estimators of these functions, and provide guarantees on the L_2 error of the resulting estimates under any given distribution of state/action values. In this analysis, they provide some theoretical results which suggest that, using novel forms of regularization targeted towards some given distribution, they can likely achieve lower L_2 error w.r.t. that distribution. Finally, they test their theory with some simple synthetic experiments.

**Questions:**

Questions/Comments below:
1. Saying that estimating the Q-function / state density ratio is “Off-Policy Evaluation” is confusing and contrary, I believe, to most of the literature. Usually, OPE refers specifically to what you call “return estimation” in section 1. I think it is worthwhile making the language use more clear in title/abstract/intro, and making it more clear that the focus of the paper is NOT on performing OPE, but rather on the related task of estimating value / weight functions, which can be used as a sub-routine for OPE (or other RL tasks).
2. $d^D$ on line 57 of the introduction is not defined yet at that point.
3. Lines 149-150: you say that “even estimating a single constraint is infeasible as that requires sampling from the same state multiple times”, but it seems like for any given (s,a) the constraints should be identifiable, since the distributions of r and s’ given (s,a) under the observational distribution should be identifiable. Therefore I’m not clear what you mean by this. This requires more detail / explanation.
4. On line 205 (and later), I assume that $f’_{s,a}$ refers to the derivative of $f_{s,a}$? This notation is currently undefined and should be made explicit.
5. In figure 2, it is not very clearly explained what is going on with the “model” data. I think based on the parantheses in the text these are showing results where we use $\tilde q$ as the estimate rather than using your regularized estimator? This should be more clear.

**Limitations:**

Issues with limitations already addressed in the strengths/weaknesses section. No negative societal impact issues that I think need to be addressed.

**Strengths And Weaknesses:**

Edit: see follow up discussion below for updates on these points

Strengths:
- The theory is very clean and easy to read / understand
- They provide a nice, more novel framework for Q-function and state density ratio estimation, with very strong theoretical guarantees for these tasks
- Their experiments are clean and well presented, and make a reasonably strong case that their novel regularization is good at reducing L_2 error under the target distribution of interest
- Despite the weaknesses described below on the impact of this work, one positive impact is that it allows a strong relaxation of the identification conditions needed by existing minimax methods for Q-function and state density ratio estimation used in the OPE literature, which is significant.

Weaknesses
- The problem of needing to obtain strong guarantees on the accuracy of Q-function and state density ratio estimates is not very well motivated. As the authors point out, the main way in which these functions are used in practice is for policy value estimation, which is already very well studied in past work with guarantees for the actual task of interest. The authors mention for example that learning these functions can be used as sub-routines for other RL tasks such as actor-critic methods, but they do not provide much/any detail about this.
- Related to and expanding on the previous point, since learning the Q-function or state density ratio is not particularly useful in and of itself, it is disappointing that they do not show show that improved performance in learning these functions can actually translate to improved performance in downstream tasks where they might be used. For example, they could provide either: (1) theoretical results that show that realizable versions of their proposed estimators can give improved guarantees for downstream tasks; or (2) empirical results that show that their proposed estimators can improve the performance of downstream tasks in practice. Without either (1) or (2), or something similar, it is not clear that the work is actually high impact.
- Similar to the above two points, it is also not very clear how useful it is to be able to guarantee low L_2 error under particular target state/action distributions. For example, in the introduction they note that “downstream learning algorithms that use off-policy function estimation as a subroutine often assume the estimation to be accurate under certain specific distributions”, but unless these are fixed, known distributions it is not clear how their theory can be utilized, since they can only provide error guarantees w.r.t. fixed, known distributions. Again, without more analysis about how their theory can actually improve downstream tasks the impact is hard to assess. A concrete and detailed example case study of the use of the theory would be very helpful with these issues.
- The results about how novel (s,a)-specific forms of regularization can improve performance is fairly heuristic. For example, they suggest using $f_{s,a}(x) = \frac{1}{2}(x - \hat q^\pi(s,a))^2$ for Q-function estimation, where $\hat q^\pi$ is some first-stage estimate of the q function, which they argue results in zero error when the first-stage estimate is perfectly accurate, but they do not provide any actual concrete bounds on the resulting L_2 error given provable bounds on the accuracy of $\hat q^\pi$. Similarly, in their experiments they do not test these novel regularization methods with actual first-stage estimation of $\hat q^\pi$, but only with artificial error added to $q^\pi$, which makes the empirical usefulness of this novel regularization difficult to assess.
- There are several trivializing assumptions made, including: (1) finite state space; and (2) finite function classes. While it is true that often extending results using finite function classes to infinite ones using e.g. VC-dimension or Rademacher complexity is straightforward, this is not necessarily always the case (or at least making this exertion is sometimes less straightforward or requires additional assumptions), so not including more general results is definitely a weakness. Similarly, is there any reason why the state space was assumed to be finite? This is an unrealistic assumption in many settings, and none of the results depend on the size of the state space, so if there are any theoretical reasons why the theory doesn’t naturally extend to infinite state spaces this should be made clear. Similarly, the iid data assumption is generally unrealistic, although I am more sympathetic there since it is usually trivial to replace e.g. concentration inequalities, CLT, etc. with corresponding Markovian versions given mixing assumptions; however, there should at least be some discussion in the paper about this.

---

> ### Author Response · Authors · 2022-07-28
> **Response**
>
> We thank the reviewer for the detailed and valuable feedback.
>
> ---
>
> - “the main way in which these functions are used in practice is for policy value estimation, which is already very well studied in past work.”
>
> We provide a detailed response in Common Response above, e.g. how off-policy function estimation is of great interest in online and offline training, with concrete motivating examples. We do not claim these functions are mainly used for policy value (return) estimation, although it is one possible application. Even for return estimation, studying function estimation may still be important: Ex. 4 in the Common Response shows how function estimation is used as the base algorithm in model selection for return estimation.
>
> ---
>
> - “they do not show that improved performance in learning these functions translate to improved performance in downstream tasks”
>
> In some of the motivating examples provided in the common response, existing works assume access to well-estimated $\hat{q}$ and $\hat{w}$ under some distribution, without showing how to achieve such guarantees. Hence, giving these guarantees in the first place is already a significant contribution. As function estimation is a concisely defined “minimal” problem and has fundamental importance in the broader RL community, we believe the results in this paper alone are sufficient for publication.
>
> While your suggestions (on showing theoretical and empirical improvements in downstream tasks) are out of the scope of our paper, they are very interesting directions for future work. Your point is well-taken and resonates with us: as we have acknowledged, there are still “gaps” between our theory and downstream tasks in some examples, and this discussion is included in our revised pdf.
>
> ---
>
> - “How novel (s,a)-specific forms of regularization can improve performance is fairly heuristic… ...$\hat q^\pi$ is some kind of first-stage estimate… which they argue results in zero error when the first-stage estimate is perfectly accurate, but they do not provide any actual concrete bounds on the resulting L_2 error given provable bounds on the accuracy of $\hat q^\pi$”
>
> A few comments are in order:
>
> 1. $\hat q^\pi$ does not have to come from a separate estimation procedure. It can be guessed from domain knowledge. Most previous works are essentially making a guess of 0, and if one has a better guess than 0 it will generally improve performance. For example, in the weight estimation case, guessing w=0 is quite bad because importance weights have expectation 1 and are not centered around 0. Moreover, in our experiments, we use a constant for the model, and show it is empirically very effective.
>
> 2. “Results in zero error when first-stage … is perfect.” Not quite. Our algorithm still suffers estimation errors when $\hat q^\pi$ is perfect,; it’s just that the optimal dual solution is $w_f^* \equiv 0$, which potentially makes learning easier.
>
> 3. Our bounds apply to arbitrary $\hat q^\pi$ as long as it does not depend on the randomness in the data used by our algorithm, so in that sense it is general enough to handle an arbitrary first-stage estimation process (assuming data splitting). That said, the tricky part is that our realizability assumption depends on $\hat q^\pi$ through $f$, so the realizability assumptions themselves become random, which can be undesirable as, generally, the function classes are fixed. This is partially mitigated by our results in Appendix E, which contains a more general analysis handling approximation errors. Though the issue remains, we don’t think there is much to do on the theory side, i.e., it seems less of a problem with our analysis, but just reflects the messiness of reality. If the reviewer has an idea how this can be easily addressed, we would like to hear.
>
> 4. While using the (s,a)-specific regularization does enjoy (partial) theoretical support, we agree it is heuristic to some extent, but this isn’t necessarily a bad thing. Strict theory can have  limited practical relevance, especially in RL where most bounds are very conservative, and extrapolating beyond the literal theory can produce the most practically useful insights.
>
> 5. We empirically mix $q^\pi$ with a constant to create a well controlled experimental setup, which we feel is a valid way to test the effectiveness of the regularizer. Note that even when the model becomes a constant (leftmost point on Fig 2(1)), it is still significantly better than no regularization (“none” in Fig 2(1)) and regularizing with 0 (e.g., compare the y-axis scale of Fig 1(3) and 2(1)).

---

> > ### Author Response · Authors · 2022-07-28
> > **Response (cont.)**
> >
> >
> > - “Saying [function estimation] is ‘OPE’ is confusing and contrary… to most of the literature.”
> >
> > Fun fact: important literature on OPE such as [JL16] has “off-policy *value* evaluation” in the title, precisely because they consider return estimation and want to distinguish it from “off-policy evaluation” which can be easily interpreted as off-policy function estimation from the broader RL community. Lihong Li and coauthors later used other terms such as “off-policy *estimation*”, again, to distinguish from “off-policy evaluation” for the same reason. It is not until quite recently when OPE becomes synonymous with return estimation within offline RL theory, but in the broader community it still means both. Plus, we clarified what we meant in the very first sentence of the abstract & introduction, so there should not be any confusion.
> >
> > ---
> >
> > - $d^D$ not introduced on L57.
> >
> > We mentioned that $d^D$ is the data distribution on L44. We will repeat this in L57 as well.
> >
> > ---
> >
> > - “L149-150… but it seems like for any given (s,a) the constraints should be identifiable… Therefore I’m not clear what you mean by this”
> >
> > We agree it’s identifiable (we didn’t say it’s not), but identifiability is an asymptotic concept. In offline RL, we are concerned with very large state spaces in the finite-sample regime (e.g., # sample sizes << # states).  The constraint for any given (s,a) cannot generally be estimated given a finite sample (of reasonable size) because, practically, we may never run into the same state twice.
> >
> > Again, we think this is a difference in culture / conventions which we touched on in the response to common concerns: in (a substantial part of) the RL theory community, the default is to think about large state spaces, limited samples, never seeing the same state twice, and generalization via function approximation of limited statistical capacities. Asymptotic identifiability is less relevant in this setup.
> >
> > ---
> >
> > - Notation for derivative $f’$ & “model” in Fig 2.
> >
> > You are correct on both. We have clarified the $f’$ notation in L205 of the revision, and added explanation of the “model” data to the caption of Fig 2. We will make this clearer in a future revision--thanks for pointing this out.

---

> > > ### Comment · Reviewer_e62P · 2022-08-09
> > > **Response to Response**
> > >
> > > Thanks for the detailed and well thought out response! A few quick points below:
> > >
> > > - First, apologies on the points about the usage of OPE in the literature and my mistake on not defining $d^D$. Admittedly my RL background is mostly on what you are calling return estimation and in that literature the problem of return estimation is almost always just called OPE, but I can see now that the language usage is different in different parts of the RL community. That being acknowledged, I still think it would be worthwhile adjusting the title and abstract more significantly to avoid confusion from the large "return estimation" community, but this is obviously a debatable point. However, thankfully both of these issues were very minor in my original assessment and barely factored into my scoring.
> > >
> > > - Regarding the motivation for the task, the revised paper with Appendix A resolves this somewhat for me. That said, these examples are all kind of vague (e.g. just saying that the estimation "needs to be accurate”). It would greatly improve the motivation if there were actual concrete examples of how the accuracy of the function estimation improves these tasks (e.g. sample complexity or PAC bounds in terms of the function estimation error, or something like this). This would give a sense of what kind of actual concrete impact improved function estimation should actually have on the downstream tasks.
> > >
> > > - RE the point about lines 149-150 I understand your point about finite sample guarantees being difficult, but what you say in the paper is “even estimating (a candidate $q$’s violation of) a single constraint is infeasible”, which is vague since you definitely  estimate these things, and provide formal guarantees on the corresponding estimation error. What you actually seem to want to say is that said estimates would have too large finite-sample errors (due to the kind of issues you mention) for them to be useful. This is of course a somewhat minor and nit-picking point, but even so it should be made more clear.
> > >
> > > - While it’s definitely true that no one conference paper can solve every aspect of a problem, as acknowledged e.g. in your common response on the motivation of the problem, there are *very* significant gaps between your proposed solution and the problem it is geared towards, both theoretically (since in your motivating examples, the distribution you need accuracy for is unknown, whereas you provide guarantees for fixed and known distributions), and empirically (since you do not empirically test what is important about your function estimates, i.e. to what extent they can improve downstream tasks, and rather you only empirically test a goal that is unimportant in and of itself, i.e. how accurate are the function estimates). Therefore, the work seems to be incremental, and the main impact of it would be in however helpful it is as a stepping stone for future works that actually seek to address these gaps.
> > >
> > > Given these considerations, I’m updating my recommendation to a weak accept, since I would now consider it to be technically sound (if very incremental) work towards solving an important problem, rather than technically sound work towards solving a poorly motivated problem. The work definitely has value, in my opinion it’s just not necessarily as high impact as a lot of the work that is typically accepted at NeurIPS.

---

> > > > ### Author Response · Authors · 2022-08-10
> > > > **Thank you for your response**
> > > >
> > > > Dear reviewer,
> > > >
> > > > We thank you very much for your detailed and sincere reply, and we very much enjoy the open and honest conversation (hope you do too). Also thanks for acknowledging that the rebuttal addresses partially your concerns on motivation, and mentioning that a few other issues are minor and do not affect your score in major ways (eg the OPE terminology, and the “identifiability” issue—we will further clarify in the paper according to your suggestions). We also respect your position on the significant “gaps” and think it’s just a difference in perspectives — to us, off-policy function estimation is such a natural problem long studied in RL, and a clean weighted L2 result itself should be of significant interest to a very broad community. But we can certainly understand your sentiment from the perspective of the OPE theory community (which we also overlap with significantly).
> > > >
> > > > There are two more things we would hope you can clarify:
> > > >
> > > > 1. Your original review mentioned finite hypothesis classes, discrete states and actions, and iid data as a significant restriction. We responded in detail how handling continuous states and iid data are straightforward and do not affect our main messages, and included a covering-based analysis of infinite hypothesis class in the revised pdf. You didn’t mention these in the comment, and could you please let us know if our response and results address your concern?
> > > >
> > > > 2. You mentioned that you increased the score to “weak accept”, but the score in the system shows as borderline accept. We totally understand that by weak accept you may not mean it in the literal sense, but thought we’d just double check to make sure this is not an error.
> > > >
> > > > thanks,
> > > >
> > > > Authors

---

> > > > > ### Comment · Reviewer_e62P · 2022-08-10
> > > > > **Clarification**
> > > > >
> > > > > You're very welcome! I agree, open dialogue is good, and I am frequently frustrated with poor review processes with opaque judgements and no engagement. Here's some clarification on the two points:
> > > > >
> > > > > 1. I'd mentioned those simplifications as a minor weakness (much more minor in my opinion than the other weaknesses I listed), and I didn't have any follow up comments on that. I definitely understand that those things like assuming finite state space or finite hypothesis class are very common in the RL literature. Personally I think that is very unfortunate, and is something that should change (generalizing these things when possible is usually very straightforward, and doesn't complicate things too much, so why not just do it... otherwise it can be difficult to tell whether these simplifications are actually fundamental in any way to the problem), but admittedly that is partly my personal opinion, and again I understand that it is common, so I wouldn't be harsh about it
> > > > >
> > > > > 2. I mistakenly clicked borderline rather than weak when I first edited my review and changed it to weak immediately after, but I'm assuming you probably got a notification about the first change. It should be good now though I think.
> > > > >
> > > > > Thanks!

---

> > > > > > ### Author Response · Authors · 2022-08-10
> > > > > > **Thanks!**
> > > > > >
> > > > > > Understood. Thanks again for carefully considering our responses. Have a good night (or morning/whatever time in your time zone)!
> > > > > >
> > > > > > best,
> > > > > >
> > > > > > Authors

---

### Official Review · Reviewer_NNnr · 2022-07-22

**Rating:** 5
**Confidence:** 3
**Soundness:** 3 good
**Presentation:** 3 good
**Contribution:** 2 fair

**Summary:**

The authors provide finite sample Lp rate guarantees for q function estimation via minimax objectives. Unlike prior work they manage to prove guarantees with only realizability assumptions. Prior works obtained such results only for policy evaluation but not q function estimation. Or required function spaces for the adversarial weights that are too expressive to leverage statistical power.


**Questions:**

it seems that many techniques are already existent in the prior work of JHHJL22. If one ignores the extension to measuring dists and just looks at RMSE, then can you elaborate on what new you are bringing on the technical side

**Limitations:**

Given that this is mainly a theoretical contribution, the above points would need to be addressed before publication.

**Strengths And Weaknesses:**

I found the theory interesting and well presented. The results could be of interest to the community of offline RL.

My main concerns are:

why would one estimate a q function other than in order to estimate an offline policy value. Please provide more motivation as otherwise the results are covered by prior work.

why are you restricting to finite action and state spaces. Given all your realizability assumptions it would seem that this is an overkill and defeats the purpose. These function approximation would be primarily most powerful with continuous states/actions.

it is very unfortunate that only slow rates are obtained. Hence even for finite hypothesis spaces one gets n^(1/4) rates. Such realizability guarantees and also the absence of ill-posedness in this inverse problem should be leading to fast rates. See for instance the un-cited work of Chen and Qi: https://arxiv.org/abs/2201.06169 Please relate to this work in your response as it is highly relevant.
why are you only giving bounds for finite hypothesis spaces. This is very restrictive.

---

> ### Author Response · Authors · 2022-07-28
> **Response**
>
> We thank the reviewer for their valuable comments. We have responded to concerns on motivations and simplification assumptions above, and will address the remaining concerns below.
>
>
> - Slow rate and comparison to https://arxiv.org/abs/2201.06169
>
> We agree that the slow rate is indeed a weakness, and we discussed the barriers to fast rate in detail in Appendix A. A key challenge in our setting is to rely only on realizability assumptions on function classes, i.e., the only property we allow on each function class (other than bounded magnitude and complexity) is that it contains a certain a priori defined function. Most prior works that obtain the fast rate, including the one reviewers mentioned, require key assumptions that are either arguably stronger (e.g. linear or Bellman-completeness) or different (e.g. matrix invertibility in LSTDQ). We are completely unaware of any technical tools for obtaining $1/\sqrt{n}$ rate with only realizability assumptions (apart from this recent paper which makes an additional strong gap assumption: https://arxiv.org/abs/2203.13935), and this has been an open problem since JHHJL22. Therefore, we are quite confused as the reviewer seems to suggest that $1/\sqrt{n}$ should be easy, and if so we would like to see pointers to relevant papers and techniques.
>
>
> > “the absence of ill-posedness in this inverse problem should be leading to fast rates… see … Chen and Qi”
>
> Thank you for pointing us to this paper, which we will cite and discuss after a more careful read. From our current understanding, Chen & Qi make a number of assumptions that we do not, detailed below, thus their rates and ours cannot be directly compared.
>
> The key assumption that enables fast rate in their paper is the “well-posedness” of Sec 3.2, i.e. $\bar\tau \lessim 1$ due to Assumption 4. This is not a common assumption considered in offline RL theory literature, as it seems to imply that the transition dynamics are nearly-deterministic. Generally, $\bar\tau\ge 1$ due to the variance from state transitions present in the numerator (as the paper points out). Near-deterministic dynamics can offer a lot of conveniences which we do have access to: for example, the infamous double-sampling difficulty is known to be not an issue in deterministic environments, and we suspect it might not matter much in near-deterministic environments either. In contrast, MIS is designed to address fully-stochastic environments and handle the double-sampling issue head-on by using the “average Bellman equations” (Eq. 2).
>
> Moreover, the main guarantee in Chen & Qi (Theorem 1) is not a complete “end-to-end” analysis like ours, and as the paper mentioned, this general analysis can be combined with other works to produce “end-to-end” results, e.g., “For example, combining our Theorem 1 with Theorem 11 of Farahmand et al. (2016)... Applying our Theorem 1 to Example 6 of Uehara et al. (2021)...”, and both referred papers require Bellman-completeness-like assumptions, which we want to avoid in our work.
>
> Sec 5 proposes the Sieve estimator, and Eq.15 looks very similar to LSTDQ. LSTDQ can be viewed as a special case of Farahmand et al & Antos et al when using linear classes (which can be analyzed under Bellman-completeness), or can be analyzed under the matrix invertibility condition [PKBK22]. Both directions are discussed in our related work section; in short, our analyses operate under weaker assumptions (realizability) so the rates cannot be directly compared.
>
> Finally, Assumption 4 makes boundedness assumptions on the absolute probabilities (p_min and p_max). It seems that $1/p_{\min} \ge |\mathcal{S}|$, so dependence on  $1/p_{\min}$ might hide dependence on $|\mathcal{S}|$, which is again, something we would like to avoid.
>
> We apologize if we misunderstand the paper given the short reading time; please correct us if the above description contains technical mistakes. But overall we believe our high-level point stands: this is a relevant paper we should discuss, but its results are not comparable to ours for reasons already discussed in the paper. The fact it gets fast rate (under several assumptions we do not make) does not imply it should be easy for us to get fast rate.
>
>
> > “Even for finite hypothesis spaces one gets n^(1/4) rates”
>
> As mentioned in the Common Response, handling infinite classes is straightforward, and the rate would be the same under standard complexity measures, so we don’t think finite classes are giving us any advantages. If anything, we’d guess that infinite class might be *easier*: we mentioned in Appendix A the work of [ZHWZ21] on fast rate for saddle-point generalization (which does not directly apply to our setting), and a key assumption in [ZHWZ21] is a Gradient Lipschitz continuity (Assumption 3), which only makes sense for continuous hypothesis classes.

---

> > ### Author Response · Authors · 2022-07-28
> > **Response (cont.)**
> >
> > - “Many techniques are already existent in the prior work of JHHJL22”
> >
> > Correct, and we openly acknowledge it on L106, with further detailed discussion in Appendix A.  However, we believe the results and insights of this paper are far from obvious, and of interest to the community, even for someone who has read JHHJL22; see Appendix A for a summary of interesting differences and comparisons between JHHJL (policy learning) and our work (OPE). A crucial one is that the optimal dual solution (the counterpart of our $w_f^*$) does not have a closed-form expression in JHHJL due to the additional complexities in policy learning, but we find that it does have a very simple form in OPE, which leads to many novel discoveries and insights in our paper. Coming up with brand new technical tools is not the only way to make interesting contributions, especially for venues like NeurIPS.

---

### Author Response · Authors · 2022-07-28
**Common Response**

We thank the reviewers for the valuable feedback and respond to the common concerns shared by reviewers e62P and NNnr. We first give a brief summary and response to each concern, then answer in more detail below.

---

-  **Motivation for function estimation**

> “learning the Q-function or state density ratio is not particularly useful in and of itself,” (e62P)

> “why … estimate a q function other than … to estimate an offline policy value.” (NnnR)

> “the main way … these functions are used in practice is for policy value estimation” (e62P)

TL;DR: While it is true that recent offline RL theory is concerned with return estimation, in the broader RL community there has been a long history of interest in off-policy function estimation. E.g., Sutton & Barto textbook almost exclusively talks about OPE in the context of function estimation instead of return estimation. To make this motivation more clear, below we thoroughly discuss a number of motivating examples. Some of them were lightly touched on in the submission, and we have included these further details in **Appendix A** of the revised pdf.

---

- **Simplification assumptions such as finite state-action spaces, finite hypothesis classes, and iid data. (e62P bullet points 1-3)**

TL;DR: Finite (but arbitrarily large) state/action spaces and iid data are standard in recent RL theory and adopted by many influential papers. They allow us to have a clean presentation accessible to a wider audience, while still capturing the essential challenges of the problem. As e62P guessed, extending our analysis beyond the finite case is indeed straightforward, and they do not change or add little to the take-away messages of the paper. **To that end, we include the analysis of infinite classes based on $\ell_\infty$ covering in the revised pdf, which follows straightforwardly from textbook techniques in statistical learning theory.**

---

> ### Author Response · Authors · 2022-07-28
> **Motivation for Function Estimation**
>
> Off-policy function estimation is one of the most fundamental questions in RL. Reviewers are invited to open the textbook of Sutton & Barto, and will see that all OPE concepts---even those like importance sampling (Section 5.5), which most of us would associate with return estimation---are introduced in the context of function estimation. Anecdotally, more RL researchers (theoretical and applied) are interested in off-policy function estimation compared to return estimation outside the offline RL theory community, as the former can be used to improve online RL (e.g. by using logged data to improve sample-efficiency).
>
> We can provide multiple concrete scenarios to motivate the function estimation task, most of which have been mentioned briefly in the submission. We have also included these details in Appendix A of the revised pdf and a pointer to it the main text. (We will also try to incorporate some of the details in the main text, if space and the flow of the main text permit, but that will require more time for careful writing and structuring.)
>
> 1. Weight learning (from Footnote 1): [LSAB19] design an off-policy policy gradient algorithm that requires estimating the density-ratio $w^\pi$ to correct the offline data distribution to the on-policy distribution. In their convergence analysis, they assume access to a blackbox $w^\pi$ estimator that is accurate under $d^D$, and refer to [LLTZ18] as a possible method. However,  as we argued around Prop 1, [LLTZ18] and existing works do not provide desirable guarantees for such a task, but we do.
>
> 2. Value-function learning (from L60): The seminal paper of [KL02] designs the CPI algorithm for on-policy policy improvement, which inspired popular empirical algorithms such as TRPO and PPO. CPI requires an oracle for estimating the advantage function (≈ value function up to offset) accurately under the on-policy distribution, i.e., distribution induced by the current policy (see their Sec 7.1). While this is easy to do by simple squared-loss regression onto on-policy trajectories, it can be sample-inefficient as it fails to leverage off-policy data collected by previous policies. On the other hand running something like TD on all data considers a distribution different from the on-policy one. Our method offers a direct solution: use all data in the Bellman error part of the objective, and only use on-policy trajectories in the regularizer.
>
> 3. Value-function learning (from L60): [AYBBLSW19] designs a no-regret policy optimization algorithm assuming access to value-function estimation oracles. In their Theorem 5.1, they assume that the oracle outputs an esetimateof $q^\pi$ that is accura under $\nu = d^{\pi^*}$. While $d^{\pi^*}$ is obviously not accessible to us and our method does not apply as-is (related to a comment from e62P; see below), one might use our theoretical insights to design heuristics, such as upweighting high-rewarding states in the offline distribution, as a way to mimic $d^{\pi^*}$.
>
> 4. Model selection in offline return estimation: Hyperparameter tuning is a huge practical hurdle in offline return estimation [Pai+20], i.e., all OPE estimators for return estimation (except for importance sampling which has exponential variance) require some form of function approximation, and it is hard to choose the right function class with offline data alone. To address this issue,  [ZJ21] proposes a model selection process over candidate function estimates of $q^\pi$, which must be provided by base algorithms that perform function estimation.
>
> All being said, we want to acknowledge a spot-on remark from Reviewer e62P, that
> > “unless [distributions that downstream tasks care about] are fixed, known distributions it is not clear how their theory can be utilized”
>
> > “more analysis about how their theory can actually improve downstream tasks [would be useful]”
>
> In the above examples, there are some gaps between what our theory can offer and what downstream tasks need, e.g., Ex. 3 needs $\nu = d^{\pi^*}$ which is unknown; Ex. 4 assumes uniformly accurate $q^\pi$ is among the candidates and it’s unclear what the right distribution $\nu$ is. We fully acknowledge these gaps, but also think these are very interesting future directions which our work has opened, and it might be unrealistic to expect merely 1 conference paper to address all these gaps. Our work has made an important step in this direction, has interesting discoveries and is self-contained, and gives novel and strong guarantees to a concisely defined problem (a.k.a. off-policy function estimation) of central importance in the broader RL community. We hope the reviewers judge the paper based on what it offers instead of what can be done in the future.

---

> ### Author Response · Authors · 2022-07-28
> **Simplification Assumptions**
>
> - Finite hypothesis class
>
> > (e62P) “often extending results … to infinite ones using e.g. VC-dimension or Rademacher complexity is straightforward, ...though this is not always the case”
>
> Extension to infinite hypothesis classes is straightforward, and we include this in Appendix G of the revision. To briefly explain why this is the case using $q$ estimation as an example ($w$ is similar): the only concentration bound in the analysis is Lemma 8; all the rest of the analysis is about how this $\epsilon_{\textrm{stat}}$ propagates to the function estimation error. One simply needs to replace Lemma 8 with a concentration bound that depends on appropriate complexity measures. In the revised pdf, we show how this is very straightforward using $\ell_\infty$-covering numbers of $\mathcal{W}$ and $\mathcal{Q}$, respectively, which are standard complexity measures and can be induced from the pseudo-dimensions of $\mathcal{W}$ and $\mathcal{Q}$. Essentially we need to consider the complexity measure of the cartesian class $\mathcal{W}\times\mathcal{Q}$ w.r.t. our loss function, which only involves arithmetic transformations of and compositions between $w$ and $q$.  $\ell_\infty$ covering is friendly to arithmetic compositions, so the analysis follows easily from textbook techniques in statistical learning theory.
>
> ---
>
> - Finite state-action spaces
>
> > (e62P) “Is there any reason why the state space was assumed to be finite? …none of the results depend on the size of the state space.”
>
> > (NNnr) “Finite state and action spaces] defeat the purpose… [of] function approximation…[which] would be most powerful with continuous states and actions.”
>
> Our results readily extend to continuous state-action spaces, at the cost of lengthy measure-theoretic notations which hurts readability to the general RL audience. This is because, as e62P points out, we do not incur any dependence on size of state space, which is made possible by function approximation. To this end, we respectfully disagree with NNnr’s comment that a finite state-action space “defeats the purpose” of function approximation.
>
> In fact, discrete but arbitrarily large state spaces (where complexities are not allowed to depend on |S|) are the default setting today in many influential RL theory papers on function approximation, even outside offline RL theory (e.g., online RL for function approximation). Below are a few examples; some do not explicitly say the word “discrete” because it is standard and therefore omitted, and you can tell from the lack of measure-theoretic notations in their papers:
>
> Jiang et al. 2017. Contextual Decision Processes with low Bellman rank are PAC-Learnable.
>
> Jin et al. 2021. Bellman Eluder Dimension: New Rich Classes of RL Problems, and Sample-Efficient Algorithms.
>
> Du et al. 2022. Bilinear Classes: A Structural Framework for Provable Generalization in RL.
>
> (We can go on forever with this list…!)
>
> All that said, we understand that RL is an interdisciplinary area, and for readers from certain backgrounds (e.g., EE & control), treating continuous spaces with appropriate measure-theoretic notations is default and conventional, and explicit reasons are required if one wants to avoid them. However, there are senior and well-respected RL theoreticians who prefer the readability and cleanness of discrete spaces, e.g., Csaba Szepesvari (https://twitter.com/CsabaSzepesvari/status/1328886152808554496).
>
> > (e62P) “[Finite states are] unrealistic”
>
> We respectfully disagree. All computers have finite memory and precision, so almost all states we run into in practice are discrete and finite (though the state space can be enormous). Continuous states are mathematical constructs and an idealization, which can be powerful in scenarios where they enable useful technical tools (e.g., how the continuous gradient flow helps us understand gradient descent, which is actually discrete). When they do not (as is the case in many recent RL theory works), it should be fine to avoid them.
>
> ---
>
> - IID data
>
> Our analysis can also be straightforwardly extended to non-iid data, by deriving the statistical error (Lemma 8) using Martingale, instead of Hoeffding, concentration inequalities. This is duly noted by e62P as well:
>
> > (e62P) “I am more sympathetic [about data iidness] since it is usually trivial to replace e.g. concentration inequalities, CLT, etc. with corresponding Markovian versions given mixing assumptions”
>
> This is exactly why most influential offline RL theory papers adopt iid data. For details on how to handle trajectory data, the following is good reference (and assumes $\beta$-mixing):
>
> Antos et al. 2008. Learning near-optimal policies with Bellman-residual minimization based fitted policy iteration and a single sample path.
>
> Again, any change to handle non-iid data will simply enter through Lemma 8, i.e., replacing it with a Martingale concentration inequality.

---

### Meta-Review · Area_Chair_eBLe · 2022-08-27

**Recommendation:** Accept
**Confidence:** Less certain

**Metareview:**

The authors provide slow rates for Q-function estimation based on minimax objectives. The contribution is technically solid, but seems somewhat incremental and even though the authors provided responses to all major reviewer concerns, there is still concern by reviewers of the applicability of their result and the incrementality.

Despite this it seems a solid contribution to the RL literature.

**Award:**

No

---

### Decision · Program_Chairs · 2022-09-14

Accept